# Advanced Statistical Analysis of 3D Kinect Data: A Comparison of the Classification Methods

**Lenka Červená** [1], **Pavel Kříž** [1,2], **Jan Kohout** [3,*], **Martin Vejvar** [3], **Ludmila Verešpejová** [4], **Karel Štícha** [3], **Jan Crha** [3], **Kateřina Trnková** [4], **Martin Chovanec** [4] **and Jan Mareš** [3]

[1] Department of Mathematics, University of Chemistry and Technology Prague, Technická 1905/5, 16628 Praha 6, Czech Republic; lenka.cervena@vscht.cz (L.Č.); pavel.kriz@vscht.cz (P.K.)

[2] Faculty of Mathematics and Physics, Charles University, Sokolovská 83, 18675 Praha 8, Czech Republic

[3] Department of Computing and Control Engineering, University of Chemistry and Technology Prague, Technická 1905/5, 16628 Praha 6, Czech Republic; martin.vejvar@vscht.cz (M.V.); karel.sticha@vscht.cz (K.Š.); jan.crha@vscht.cz (J.C.); jan.mares@vscht.cz (J.M.)

[4] Department of Otorhinolaryngology, Faculty Hospital Královské Vinohrady, Šrobárova 1150/50, 10034 Praha 10, Czech Republic; ludmila.verespejova@fnkv.cz (L.V.); katerina.trnkova@fnkv.cz (K.T.); martin.chovanec@fnkv.cz (M.C.)

[*] Correspondence: jan.kohout@vscht.cz

**Abstract:** This paper focuses on the statistical analysis of mimetic muscle rehabilitation after head and neck surgery causing facial paresis in patients after head and neck surgery. Our work deals with an evaluation problem of mimetic muscle rehabilitation that is observed by a Kinect stereo-vision camera. After a specific brain surgery, patients are often affected by face palsy, and rehabilitation to renew mimetic muscle innervation takes several months. It is important to be able to observe the rehabilitation process in an objective way. The most commonly used House–Brackmann (HB) scale is based on the clinician's subjective opinion. This paper compares different methods of supervised learning classification that should be independent of the clinician's opinion. We compare a parametric model (based on logistic regression), non-parametric model (based on random forests), and neural networks. The classification problem that we have studied combines a limited dataset (it contains only 122 measurements of 93 patients) of complex observations (each measurement consists of a collection of time curves) with an ordinal response variable. To balance the frequencies of the considered classes in our data set, we reclassified the samples from HB4 to HB3 and HB5 to HB6—it means that only four HB grades are used for classification algorithm. The parametric statistical model was found to be the most suitable thanks to its stability, tractability, and reasonable performance in terms of both accuracy and precision.

**Keywords:** rehabilitation; House–Brackmann scale; functional data analysis; ordinal classification; Kinect evaluation

## 1. Introduction

Image analysis in medicine has been a popular topic in recent decades, and it is still evolving and finding further applications. Imaging techniques such as X-rays, computed tomography, and magnetic resonance imaging have been known and used for many years. However, new imaging methods have recently been developed (e.g., depth maps, 3D reconstructions, etc.) and their use in modern medicine is breathtaking.

Thus, a new area of image analysis in medicine is beginning, which is expected to be useful for diagnosis, rehabilitation, or validation of the results. This area does not aim to replace clinicians (this would not even be possible), but it does aim to save their time and help to interpret the results. A number of specific applications for biomedical image analysis can be found in the literature; for example, image analysis can be used to diagnose

a large number of neurological diseases [1,2] or sleep apnea [3,4]. Furthermore, movement analysis has been used in rehabilitation [5].

It is possible to find a number of innovative diagnostic procedures based on image analysis. A research group based in Oklahoma University introduced a radiomics-based machine learning model to predict peritoneal metastasis in gastric cancer patients using a small and imbalanced computed tomography (CT) image dataset [6]. Meanwhile, 3D reconstruction also plays an important role in precise onco-surgery [7].

This paper is focused on 3D reconstruction and statistical analysis of mimetic muscle rehabilitation after head and neck surgery has caused facial paresis. The muscles of the face include all mimetic muscles innervated by the cranial nerve VII (facial nerve). Within the parotid gland, the facial nerve terminates by bifurcating into five motor branches. These innervate the muscles: temporal branch—innervates the frontalis, orbicularis oculi, and corrugator supercilii; zygomatic branch—innervates the orbicularis oculi; buccal branch—innervates the orbicularis oris, buccinator and zygomaticus muscles; marginal mandibular branch—innervates the mentalis muscle; cervical branch—innervates the platysma. Two masticatory muscles (masseter, temporalis) that are supplied by the motoric portion of the cranial nerve V3 (mandibular nerve) also contribute to the contour of the face [8].

In our analysis, we treat the measurements (i.e., trajectories of selected points on a face) as functional data. Functional data analysis has been applied in many diverse areas, including (bio)medical data. Let us recall the earlier review paper [9] and a more recent paper [10], to name just a few. Besides the parametric model, we also applied a nonparametric approach that is based on (ordinal) random forests. A very recent application of random forests for clinical risk prediction based on complex data can be found in [11].

Clinicians require an objective, reliable, and valid clinical tool to accurately describe a patient's facial function, to monitor status over time, and to assess the course of recovery and the effects of treatment. For a grading system to have clinical usefulness, it must be easy to administer, require little time or equipment, and be sensitive enough to detect clinically important changes [12].

This work builds on previous publications and the work of several research groups dealing with similar topics [13].

### 1.1. Biomedical Background

Facial nerve palsy has a significant physical and emotional toll on affected individuals. A thorough history and physical examination are needed to narrow the broad differential diagnosis and to determine an appropriate management plan [14].

We distinguish two major types of facial nerve palsy: central (upper) motoneuron lesion between cortex and nuclei of the facial nerve in the brainstem and peripheral (lower) motoneuron lesion between nuclei in the brainstem and peripheral organs. The most common cause of lover motoneuron lesion is idiopathic facial nerve palsy, also known as Bell's palsy. Closely following Bell's palsy are infection and inflammation. Trauma, including surgical trauma in head and neck surgery, is the third most common cause of facial nerve paralysis in the general population. Other important etiologies of facial nerve dysfunction include herpes zoster oticus and neoplasms of the parotid gland, brain, and the petrous part of the temporal bone [15].

Cases of complete paralysis after surgery in which the onset of paralysis is indeterminate should be treated as immediate in nature. Delayed paralysis or incomplete paresis should be treated medically, with high-dose steroids. A good prognosis should be anticipated in these cases [16].

### 1.1.1. Electrodiagnostics of Facial Nerve Palsy

Electrophysiological tests are mainly used to determine the severity and prognosis of a peripheral facial nerve lesion [17].

Electroneurography objectively records the amplitude of electrically evoked muscle action potentials. It analyses the evoked compound muscle action potential (CMAP) of a specific facial muscle after transcutaneous stimulation of the main trunk of the facial nerve [18]. The main trunk is stimulated supramaximally at its exit from the stylomastoid foramen with a bipolar stimulator or stimulating electrodes. The CMAP is recorded using a bipolar pair of surface electrodes placed on the target muscle. Important test between 72 h and 21 days after onset, interpretation of result in comparison to needle electromyography (nEMG) result [19]. Nerve injury is expressed as percentage of function relative to the normal side [16].

Electromyography (EMG) measures volutional responses of the facial muscles without electrostimulation. A facial motoneuron unit consists of a facial motoneuron and all muscle fibers innervated by this motoneuron. Needle EMG (nEMG) is the method used to analyse a facial motor unit action potential (MUAP) recorded from a needle electrode inserted in the facial muscle. This examination is important 2–3 weeks after onset of the palsy, because pathologic activity can occur in case of facial nerve degeneration. In the later time course, nEMG is important to detect reinnervation potentials as signs of reinnervation of the facial muscles [18]. Surface electromyography (sEMG) works with voluntary activity of the facial muscles and not with external stimulation. The recording field is more superficial and larger than when using nEMG. sEMG is not used for prognostication. Multichannel sEMG is important if the interplay of different facial muscles should be analysed.

Blink-reflex testing is a test that allows stimulation of the facial nerve proximal to the lesion site. Testing is electrostimulation of the supraorbital branch of the trigeminal nerve (V1) and simultaneous sEMG recording from the orbicularis oculi muscle on both sides. Standard blink testing involves electrical stimulation of the supraorbital nerve on the affected side combined with a 2-channel simultaneous sEMG recording from both orbicularis oculi muscles. The exit of the supraorbital nerve in the supraorbital foramen is palpated on the rim of the orbit [20]. It may be most helpful if facial nerve damage is suspected to occur within the brainstem [18].

### 1.1.2. Facial Rehabilitation

Timely intervention is needed to provide patients with the best chance for recovery of facial nerve function [14].

The pharmacological treatment depends on the cause of facial nerve paresis and generally most often are used corticosteroids, antiviral agents, calcium channel blockers, vitamins to support regeneration of the nerve.

Neuromuscular electrostimulation therapy is used for the direct or indirect therapeutic stimulation of nerves, muscles. In general, three stimulation frequency ranges can be distinguished: low-frequency, medium-frequency, and high-frequency currents. The low-frequency stimulation currents (up to about 1000 Hz) are suitable for creating synchronous muscle contractions. Stimulation can be sensory or motor, but it should always be below the discomfort threshold. The current is applied via electrodes. These can either be inserted into the tissue surrounding a nerve, or into the muscle (percutaneous stimulation) or into the skin applied (transcutaneous stimulation) [21].

One of the most important parts of treatment is facial rehabilitation, which typically includes five main components, as follows:

1. Patient education to explain the pathologic condition and set realistic goals;
2. Soft tissue mobilisation to address facial muscle tightness and edema;
3. Functional retraining to improve oral competence;
4. Facial expression retraining, including stretching exercises;
5. Synkinesis management [22].

Synkinesis refers to the abnormal, unwanted, involuntary facial movement that occurs coupled with purposeful facial movement. For example, oro-ocular synkinesis occurs when movement of the lips results in a closure of the eyelids. Mild forms of synkinesis may go undetected, but severe forms cannot be ignored because they may cause severe facial

pain and facial tightness. The treatment of facial synkinesis is one of the most challenging aspects of facial paralysis care [14].

### 1.1.3. Evaluation of Facial Nerve Function

For clinicians, it is essential to evaluate the function of the facial nerve objectively. In everyday practice, the most commonly used tool is the standard grading system, such as the **House–Brackmann scale**. This system involves a six-point scale (with I being normal and VI total) of flaccid paralysis (Table 1). Group I represents normal facial movement with no weakness or synkinesis. A patient placed in group II has only slight asymmetry of facial movements with a possible slight synkinesis. Patients in group III have an obvious asymmetry with obvious secondary defects but some forehead movement. The presence of forehead movement indicates that there has not been total degeneration of the nerve. Patients in group IV have an obvious asymmetry, no forehead movement, and weakness with possible disfiguring synkinesis or mass action. When there is only slight movement of the face, no forehead movement, and not enough facial function to return to have secondary defects, the patient is placed into group V. The absence of any movement or tone places the patient in group VI [23].

**Table 1.** House–Brackmann scale (HB).

| Grade | Description | Characteristic |
| --- | --- | --- |
| I | Normal function | normal facial function in all areas |
| II | Mild dysfunction | Gross: slight weakness on close inspection; very slight synkinesis<br>At rest: normal tone and symmetry<br>Motion Forehead: moderate to good function<br>Eye: complete closure with minimum effort<br>Mouth: slight asymmetry |
| III | Moderate dysfunction | Gross: obvious but not disfiguring<br>difference between two sides; noticeable synkinesis<br>At rest: normal tone and symmetry<br>Motion Forehead: slight to moderate movement<br>Eye: complete closure with effort<br>Mouth: slightly weak with maximum effort |
| IV | Moderately severe dysfunction | Gross: obvious weakness and disfiguring asymmetry<br>At rest: normal tone and symmetry<br>Motion Forehead: none<br>Eye: incomplete closure<br>Mouth: asymmetric with maximum effort |
| V | Severe dysfunction | Gross: only barely perceptible motion<br>At rest: asymmetry<br>Motion Forehead: none<br>Eye: incomplete closure<br>Mouth: slight movement |
| VI | Total paralysis | no movement |

The House–Brackmann scale produces comparable results between different observers in patients with normal or only mildly impaired facial nerve function. Interobserver variability can increase depending on the severity of facial nerve paresis. However, House–Brackmann does not promote uniformity of reporting and comparison of outcomes in patients with moderate or severe facial nerve paresis [24].

## 2. Materials and Methods

Our study included 93 patients, with 122 undergoing head and neck surgical procedures with the specific risk of postoperative facial nerve dysfunction. The data were collected from 3.2.2019 to 14.5.2020.

Every patient was measured over a defined schedule of check-ups (the first one before the surgery and then repetitively based on a defined schedule). The patient is asked to perform a series of exercises using the mimetic muscles during the examination,

and the clinician evaluates this exercise numerically. The main disadvantage of this method is that these measurements are strongly subjective.

Figure 1 shows scheme of the whole process from data acquisition to model development.

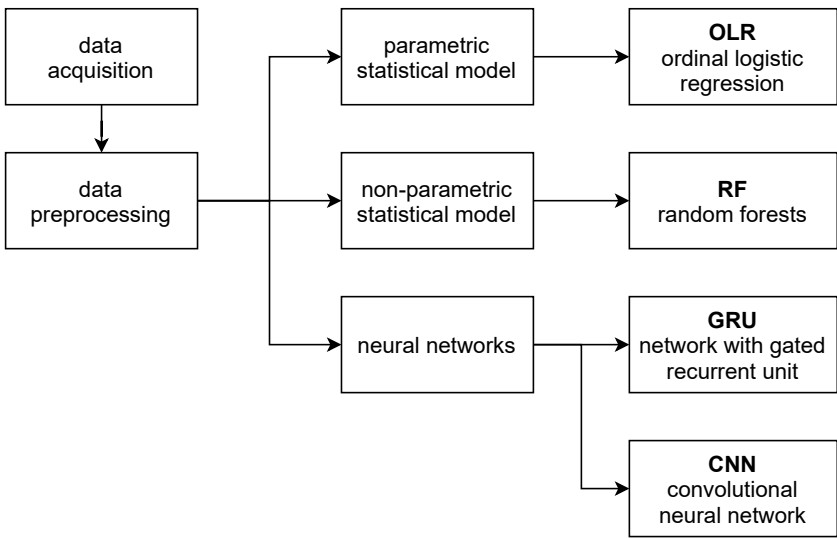

**Figure 1.** Scheme of data analysis and models.

## 2.1. Data Acquisition

A mobile robotic system and a special software developed at UCT Prague were used for data acquisition Figure 2. This system can operate in a static mode for sensing mimetics (this issue) or in a dynamic mode for sensing gait and body movements (another project solved at UCT Prague).

Kinect v2 and RealSense are used for image acquisition. The final version of this robot uses only RealSense D435 manufactured by Intel. The Kinect sensor had to be replaced because Microsoft discontinued production in 2018 and subsequently support ceased at that time without the possibility of replacement.

The robot has undergone several versions during its development over the past months, see Figure 2. Several control loops are located in the robot, which control the robot in the correct direction and distance from the patient. The basic loop is located directly in the controllers of both axles and it regulates the speed of rotation of individual wheels, which it obtains from the superior processor. The master processor then processes the odometric data from both axles and runs another loop in it, which controls the total distance travelled by all wheels based on a comparison with the theoretical position obtained from the robot model.

The robot includes adaptive cruise control, which responds to changes in the distance of the patient from the robot and to the desired position and direction of the robot in the recording corridor. The direction and position in the corridor is obtained from data from the lidar, the distance of the patient from the robot is obtained from both the lidar and the stereo camera. Consequently, in the case of failure of one of the sensors, it is possible to control the system without interruption.

During the recording, the robot stores all of the data that could be needed for a retrospective analysis of both the patient's gait or mimics and any robot failure. Currently, the system can store all of the operating data, as follows: voltage on individual cells of both LiPol batteries, odometry of each wheel, raw data from lidar, stereoscopic recording of the patient's gait, and video recording of gait.

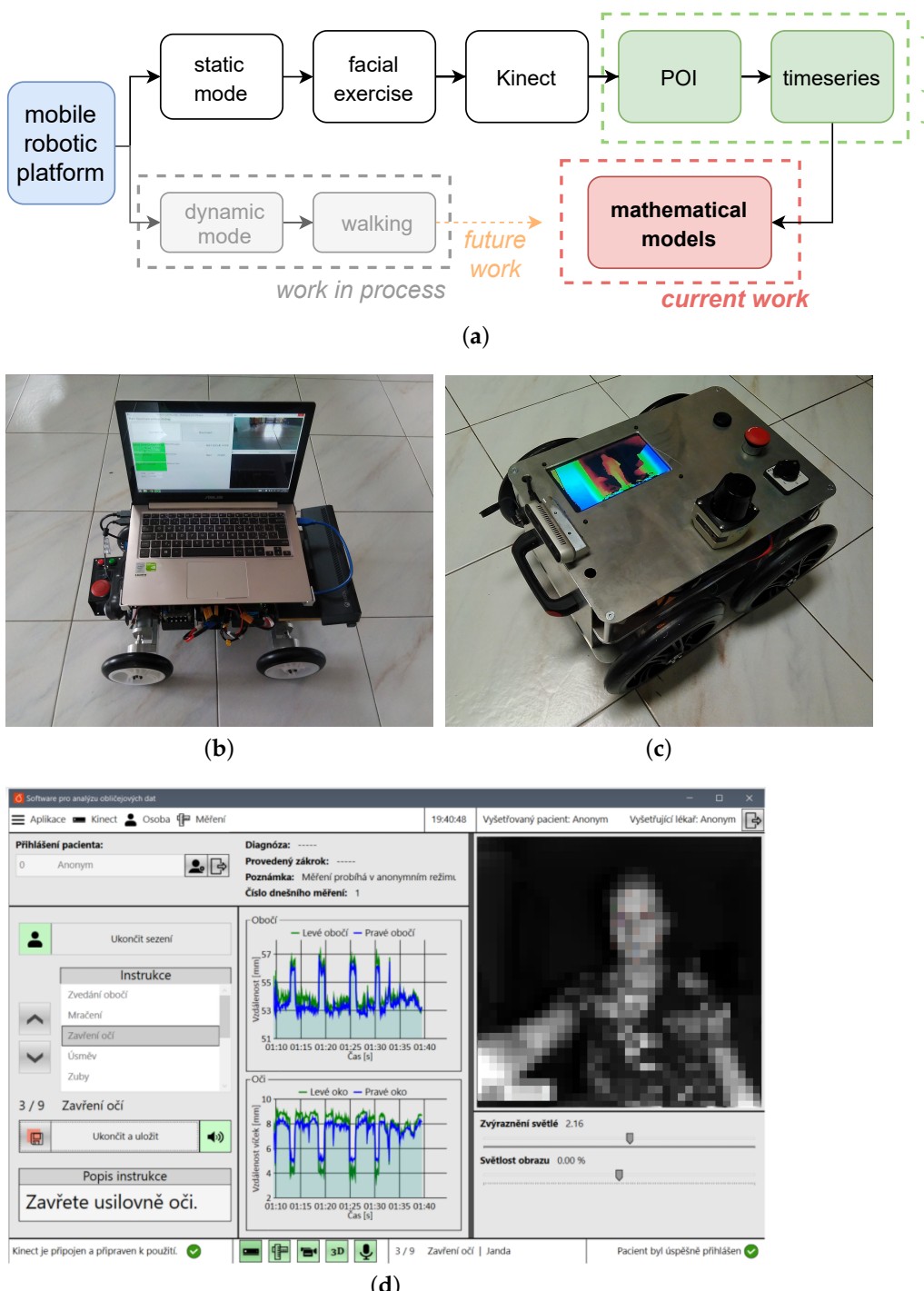

**Figure 2.** Scheme and an equipment for face rehabilitation measurement. Patient is measured during facial exercises in a hospital during regular check-ups. (**a**) Scheme of face measurement, (**b**) Mobile robotic platform used for face measuring (in static mode), (**c**) Last robot version for gait analysis, (**d**) A desktop application for data collection (face is in this image pixelized).

### 2.2. Data Preprocessing

The whole process of data preprocessing is thoroughly described in a recent paper [13]. To make this paper self-contained, let us briefly review this process here.

While being recorded by Kinect, the patients conduct their exercises at different times and at different speeds. Consequently, a temporal registration of the data is necessary for further statistical analysis. For that purpose, the original curve $P_i(t)$ representing

movement of any facial point for measurement (patient) $i$ is transformed using a time warping function $w_i$ in such a way that curves $\widetilde{P_i}$ are aligned for all $i$.

$$\widetilde{P_i}(\tilde{t}) = P_i(w_i(\tilde{t})), \text{ where } \tilde{t} = w_i^{-1}(t). \tag{1}$$

The warping functions $w_i$ are computed by a landmark registration technique, where the beginnings and end of each repetition are identified and a piecewise cubic interpolation is used for the warping function. A more detailed description of the used registration technique can be found in [13].

In the clinician's House–Brackmann  evaluation, classes HB4 and HB5 are seldom used compared to others due to their similarity with HB3 and HB6, respectively. To balance the frequencies of the considered classes, we reclassified the samples from HB4 to HB3 and HB5 to HB6.

Some exercises exhibited insufficient range of motion detectable by Kinect. In the following analysis, we consider these exercises: raising eyebrows, frowning, smiling, baring teeth, and pursing lips.

To reduce the amount of data obtained by Kinect and to access only the important behaviour, we chose several points of interest (POI) for the selected exercises and we computed various indicators from the mutual position of the selected POI at each time instant. The resulting indicator curves measure symmetry (movements in the left-half versus movements in the right-half of the face), intensity (the range of motion during the exercise), and speed (how fast the selected exercise is performed, measured by the warping function). The exact definition of all computed indicators is described in [13].

### 2.3. Compared Statistics

On one hand, we deal with the classification problem with very complex explanatory data (collection of curves for each observation) having unclear functional link to the response variable, which suggests using non-parametric statistics (or neural networks) as a preferred option. On the other hand, the number of observations is very limited, which increases the risk of overfitting for these methods. Therefore, we decided to compare parametric, non-parametric statistics and neural networks, which is less prone to overfitting, but requires very careful model specification to get reasonable results.

### 2.4. Parametric Statistical Model

Detailed description of the parametric statistical model for HB classification can be found in [13]. Let us recall that within this approach, the curves of the indicators are analysed by two parametric statistical models in two steps:

- The first step consists of application of functional logistic regression (FLR) to indicator curves (understood as functional data) separately for each indicator type (exercise). This turns the indicator curves (functional data) into health scores (real-valued data between 0 and 1).
- In the second step, classification of the set of health scores for each patient (multivariate data) into HB grades is performed using multivariate ordinal logistic regression (OLR).

2.4.1. Functional Logistic Regression

Logistic regression is a popular model that comes from the methodology of generalised linear models (GLM), which is characterised by Bernoulli-type error distribution (binary output variable $Y$) and logit link function. Functional logistic regression is a generalisation of this methodology to functional input data (functional covariates). A functional datum (a curve) is linearly transformed into real-valued linear predictor (by taking scalar product in $L^2$ space). The linear predictor is then turned into probability of a positive outcome $P(Y = 1)$ by application of the inverse link function. The corresponding formula is

$$p_k = \frac{1}{1 + \exp\left(-\alpha - \int X_k(t)\beta(t)dt\right)}. \tag{2}$$

In our setting, $X_k$ represents the curve of the particular indicator in the $k$-th sample (a functional covariate) and $p_k$ is the predicted probability of the $k$-th sample being healthy (i.e., having HB 1). We interpret these $p_k$ as health scores ("rate of healthiness") for a specific indicator and we then use them in the next step as covariates for HB classification. The parameters of the model include the scalar intercept $\alpha$ and the functional parameter $\beta$, both are estimated from the data by maximum likelihood approach.

More details about the functional approach can be found in [25]. For our calculations, we made use of an implementation of FLR in the statistical software package R (function classif.glm within the fda.usc package), see [26] for more details.

### 2.4.2. Multivariate Ordinal Logistic Regression

Given that we face the problem of classification with multiple, ordered, classes (HB grades) on the basis of the vector of health scores, we have chosen the apply the multivariate Ordinal Logistic Regression (OLR). This is a parametric statistical tool that is derived from the classical logistic regression. Whereas the classical logistic regression requires binary explanatory variable, OLR makes it possible to model ordinal explanatory variable (multiple classes with ordering). In particular, OLR applies a series of classical logistic regressions to cumulative probabilities to estimate the ordinal version of cumulative distribution function of the explanatory variable. Probabilities of individual classes (values of explanatory variable) are then predicted by taking the differences of the cumulative probabilities.

In our framework, the explanatory variable represents *HB* evaluation with ordered *HB* grades: $HB1 < HB2 < HB3 < HB6$. Regression models for cumulative probabilities take the form:

$$P(HB_k \leq j) = \frac{1}{1 + \exp\left(-\alpha_j - \sum_i \beta_i p_{ik}\right)}. \tag{3}$$

The output $P(HB_k \leq j)$ is the probability that the $k$-th sample has *HB* grade $j$ or better, $\alpha_j$ is the (*HB* specific) intercept parameter, $\beta_i$ is the coefficient for the health score of the $i$-th indicator (independent of the *HB* level) and $p_{ik}$ is the health score of the $i$-th indicator and the $k$-th sample. The parameters $\alpha_j$ and $\beta_i$ are estimated from the data by maximum likelihood method.

To avoid overfitting and to provide stable model predictions, we performed a stepwise variable selection procedure based on minimising the AIC (considering both directions, starting from the empty model).

To compensate for the imbalance of the input data, we applied weighted version of multivariate OLR, where the weights of samples were set as the inverse value of the frequency of the corresponding class.

The probability of a sample's having a specific *HB* grade is calculated as the difference of cumulative probabilities:

$$P(HB_k = j) = P(HB_k \leq j) - P(HB_k \leq j - 1). \tag{4}$$

The resulting predicted class for a sample is determined as the class with the highest probability.

More details of this method can be found in [27] (which are referred to there as a cumulative logit model). We performed our calculations of OLR using the polr function from the MASS package in R (see [28] for further details).

### 2.5. Non-Parametric Statistical Model

In this subsection, we describe the methodology of non-parametric statistical model for our problem. Non-parametric models are based on different philosophy compared to the parametric approach that was described earlier. Instead of specifying the types of regression functions a prior and calibrating their parameters from the data, the form of the functional dependence of the response variables on the covariates is constructed directly on the data. In general, this approach is typically less sensitive to model misspecification

(compared to parametric approach) but requires more data to provide stable prediction. This is the reason why we have chosen this approach as an alternative for comparison of the parametric approach.

Similarly to the parametric approach, we split the modeling process into two steps:

- In the first step, we apply kernel functional classification to turn curves of individual indicators into real-valued health scores.
- In the second step, HB grades are predicted from the lists (vectors) of health scores using the ordinal forests method.

Splitting the modeling process into these two steps is advantageous for potential further use by practitioners because it makes the whole process more tractable and it enables the practitioners to see the performance of the patient in each exercise, which underlies the final HB classification; see also [13].

### 2.5.1. Kernel Functional Classification

Kernel classification is a very popular technique for non-parametric supervised classification because of its flexibility: it generalises the concept of k-nearest neighbour. The functional version of the kernel classification is discussed in detail in the well-known and frequently cited book [29]. The main idea is to determine posterior probabilities of the groups for a new observation (new curve) according to its proximities to curves in the training dataset with a known group. The proximities are calculated via a kernel (denote $K$) is applied to a distance between curves ($d$) with respect to a bandwidth ($h$).

Although functional kernel classification can be used for the multiple classes problem, we use it for prediction of a probability of class *HB* 1 (the health score), for the reasons described earlier. In particular, we have

$$p_N = \frac{\sum_{i=1}^{n} 1_{[HB_i=1]} K(h^{-1} d(X_N, X_i))}{\sum_{i=1}^{n} K(h^{-1} d(X_N, X_i))}, \tag{5}$$

where $p_N$ is the estimated posterior probability that new observation (new indicator curve) $X_N$ has $HB1$, $X_i$ for $i = 1, \ldots n$ are indicator curves in training set with known *HB* grade denoted as $HB_i$, the indicator function $1_{[HB_i=1]}$ equals one if $HB_i = 1$ and is zero otherwise, $K$ is normal (Gaussian) kernel, $h$ is the bandwidth optimised in order to minimise loss function (misclassification rate) on the training set and $d$ is the $L^2$ distance defined as

$$d(X_N, X_i) = \sqrt{\int (X_N(t) - X_i(t))^2 dt}.$$

We calculate the health scores $p_N$ for each indicator (exercise) and both for observations in the training set (for the purpose of calibration of ordinal forests in next step) and in the testing set (for the purpose of final classification by the calibrated ordinal forests).

Our calculations were implemented in the statistical software R, while kernel classification was done via the function `classif.kernel` within the `fda.usc` package); see [26] for more details.

### 2.5.2. Ordinal Forests

The ordinal forest is an efficient non-parametric tool based on random forests (RF), which is tailored to construct a classification rule for problems with multivariate input data (both low-dimensional and high-dimensional) and with ordinal response variable. This technique has recently been introduced in [30] and it fits perfectly with our need to construct a non-parametric classifier for ordered *HB* grades from lists of health scores.

The basic idea of ordinal forests is to replace the ordinal response variable $Y$ by a latent continuous variable $Y^*$ so that $Y$ depends on (and is uniquely determined by) $Y^*$ via a non-decreasing step function. In our setting, this means that individual HB grades correspond to adjacent intervals of $Y^*$. The values of $Y^*$ (and its intervals) are optimised with the aim

of maximising the prediction performance on the training set. We then grow a regression forest with $Y^*$ as a continuous response variable.

For a new observation, we start with prediction of $Y^*$ from the regression forest and we then determine the corresponding $Y$ (*HB* grade in our case).

For our calculations related to ordinal forests, we utilised the software R package; namely, the `ordfor` function (implemented within the `ordinalForest` package) with default values of hyperparameters.

### 2.6. Neural Networks

Two deep neural network (DNN) structures were trained on the given data. The core of the first DNN structure, which is further referred to as GRU DNN, are three recurrent layers with Gated Recurrent Unit [31] cell architecture (for full pipeline see Figure 3). Given the nature of the data, the main strength of GRU DNNs is the ability to infer temporal patterns and dependencies in the time-series, if any are present.

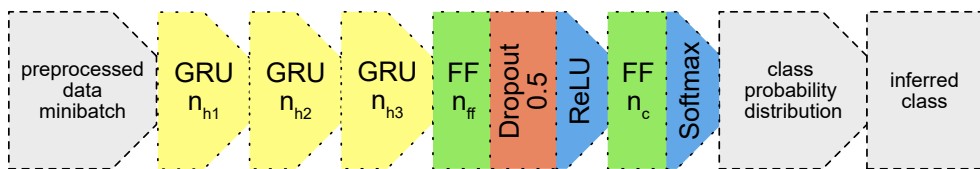

**Figure 3.** GRU DNN with three RNN layers with gated recurrent unit architectures at its core. FF marks a typical fully-connected (dense) layer. $n$ is number of hidden units in the respective layers.

The second DNN structure, which is further referred to as CNN DNN, utilises three two-dimensional convolutional layers (for full pipeline see Figure 4). Convolutional layers have a strong history in feature extraction and pattern recognition [32,33], both of which are expected of the data. They also allow for an effective dimensionality reduction and, thus, computational efficiency.

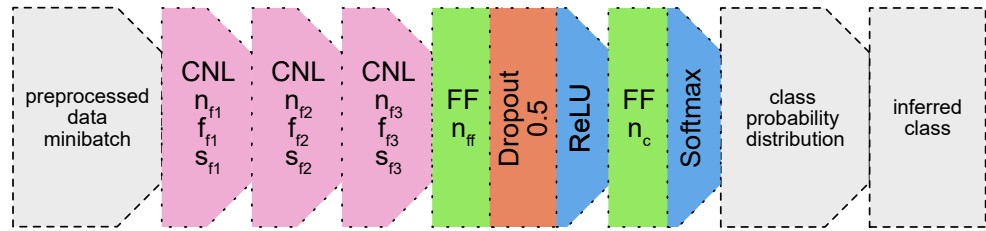

**Figure 4.** CNN DNN with three 2D convolutional layers (CNL) at its core. FF marks a typical fully-connected (dense) layer. $n$ is number of filters or hidden units, $f$ is 2D filter dimensions and $s$ is stride of the filter for the respective layer.

Both DNN structures have been trained and evaluated on the same data as the statistical methods. The parameters were optimised by minimising cross-entropy loss function using the Adam optimiser. Accuracy and F1-score, tracking both precision and recall, were used to measure performance on the validation data. For more information about performance metrics, we refer the reader to [34].

To account for imbalance in class distribution of the training data, the loss function was scaled by class weights, which were calculated based on the ratio of samples in each class.

To counteract overfitting, which was expected given the small size of the dataset, a static dropout rate of 0.5 was applied after the feed-forward layer directly following the core structure (see Figures 3 and 4). The training process was also stopped early if the average value of area under curve (AUC) on the validation data did not improve for 20 subsequent epochs and only the best performing parameters were saved.

The cross-validation results are categorised by the DNN structure type (GRU or CNN) and an indicative number of trainable parameters of the whole structure.

## 3. Results

The overall classification results of parametric OLR model and non-parametric random forest (RF) model are summarised in Table 2. The neural network classifications for selected structures are shown in Tables 3 and 4.

**Table 2.** Confusion matrix with numbers of (mis)classified cases by OLR model and random forest (RF) on a test set.

| HB by a Clinician | HB by OLR | | | | HB by RF | | | |
|---|---|---|---|---|---|---|---|---|
| | 1 | 2 | 3 | 6 | 1 | 2 | 3 | 6 |
| 1 | 28 | 17 | 10 | 3 | 48 | 1 | 9 | 0 |
| 2 | 6 | 13 | 2 | 0 | 17 | 2 | 2 | 0 |
| 3 | 2 | 3 | 15 | 3 | 13 | 1 | 8 | 1 |
| 6 | 0 | 1 | 5 | 14 | 3 | 0 | 12 | 5 |

**Table 3.** Confusion matrix with numbers of (mis)classified cases by GRU neural network structures on a test set (tp marks the number of trainable parameters).

| HB by a Clinician | HB by GRU-tp100k | | | | HB by GRU-tp1.6M | | | |
|---|---|---|---|---|---|---|---|---|
| | 1 | 2 | 3 | 6 | 1 | 2 | 3 | 6 |
| 1 | 38 | 7 | 7 | 6 | 37 | 7 | 7 | 7 |
| 2 | 9 | 7 | 2 | 3 | 6 | 11 | 2 | 2 |
| 3 | 7 | 2 | 9 | 5 | 10 | 4 | 5 | 4 |
| 6 | 5 | 2 | 5 | 8 | 9 | 4 | 4 | 3 |

**Table 4.** Confusion matrix with numbers of (mis) classified cases by CNN neural network structures on a test set (tp marks the number of trainable parameters).

| HB by a Clinician | HB by CNN-tp160k | | | | HB by CNN-tp4M | | | |
|---|---|---|---|---|---|---|---|---|
| | 1 | 2 | 3 | 6 | 1 | 2 | 3 | 6 |
| 1 | 43 | 1 | 8 | 6 | 43 | 4 | 11 | 0 |
| 2 | 8 | 6 | 4 | 3 | 9 | 9 | 3 | 0 |
| 3 | 7 | 4 | 12 | 0 | 8 | 2 | 12 | 1 |
| 6 | 5 | 1 | 3 | 11 | 4 | 1 | 1 | 14 |

Given that our goal is not to precisely predict clinician's HB scores because they may be subjective, we consider both the correct classification of HB classes (model HB equals to clinician's HB) and the approximate classification (we tolerate the model HB to differ by at most 1 from the clinician's HB class). The accuracy (or recall) of individual HB classes and the overall accuracy are depicted in Table 5. By accuracy of an HB class, we denote the fraction of correctly classified samples by the model out of all samples in that HB class (assigned by a clinician). The overall accuracy is the fraction of correctly classified samples in all classes out of all samples.

To complete the evaluation of the multi-class classification, Table 6 shows precision calculated for all HB classes individually. The precision of an HB class represents the probability that a sample given the HB class by a model is assigned the same class by a clinician.

**Table 5.** Comparison of classification of all models. Both the correct and approximate classifications are considered.

| Correct Classification | | | | | | |
|---|---|---|---|---|---|---|
| **HB by a Clinician** | OLR | RF | GRU-tp100k | GRU-tp1.6M | CNN-tp160k | CNN-tp4M |
| 1 | 48% | 83% | 66% | 64% | 74% | 74% |
| 2 | 62% | 10% | 33% | 52% | 29% | 43% |
| 3 | 65% | 35% | 39% | 22% | 52% | 52% |
| 6 | 70% | 25% | 40% | 15% | 55% | 70% |
| **Overall accuracy** | 57% | 52% | 51% | 46% | 59% | 64% |
| Approximate classification | | | | | | |
| **HB by a clinician** | OLR | RF | GRU-tp100k | GRU-tp1.6M | CNN-tp160k | CNN-tp4M |
| **1** | 78% | 84% | 78% | 76% | 76% | 81% |
| **2** | 100% | 100% | 86% | 90% | 86% | 100% |
| **3** | 78% | 39% | 48% | 39% | 70% | 61% |
| **6** | 70% | 25% | 40% | 15% | 55% | 70% |
| **Overall accuracy** | 80% | 69% | 67% | 61% | 73% | 79% |

**Table 6.** Comparison of precision of classification of all models.

| **HB by a Model** | OLR | RF | GRU-tp100k | GRU-tp1.6M | CNN-tp160k | CNN-tp4M |
|---|---|---|---|---|---|---|
| 1 | 78% | 59% | 64% | 60% | 68% | 67% |
| 2 | 38% | 50% | 39% | 42% | 50% | 56% |
| 3 | 47% | 26% | 39% | 28% | 44% | 44% |
| 6 | 70% | 83% | 36% | 19% | 55% | 93% |

The parametric OLR model achieves an acceptable accuracy of exact classification and a sufficient accuracy of approximate classification, displaying only 20% of misclassified samples (predicted HB class differs by more than 1 from the clinician's evaluation). The rates indicate balanced accuracy for all four HB classes, not preferring the dominant HB1 class. Meanwhile, the correct accuracy of the individual HB classes is not consistent in the non-parametric model random forests because the model mainly predicts the dominant HB1 class. There is also much higher number of misclassified cases (31%).

The accuracy of classification by neural networks differs substantially based on the architecture and number of trainable parameters. The GRU architecture provides sub-optimal results for both low and high number of parameters (see Table 5) with as much as 33% and 39% misclassified cases, respectively. In contrast, CNN based architectures achieved accuracies that are comparable to the parametric OLR model and benefit from higher amount of trainable parameters. The small (160 thousand parameters) CNN misclassified in 27% of test cases, while the large (4 million parameters) in only 21%. It is important to note that the number of trainable parameters of the presented neural network models

is substantially higher when compared to the size of the training set, which makes them highly susceptible to overfitting.

A comparison of the overall accuracy on the testing set with accuracy on the training datasets (for each step of cross-validation procedure) reveals that the OLR model provides stable results and does not suffer from overfitting, see Table 7. Meanwhile, the non-parametric model random forests displays overfitting: it achieves worse accuracy for testing set compared to accuracy on training sets. Non-parametric methods usually require (by their nature) rich training dataset to learn the systematic patterns and filter out non-systematic (random) noise. Otherwise, due to their flexibility, they tend to adapt to non-systematic random effects.

**Table 7.** Overall exact accuracy of classification: comparison of results on the testing set and training sets.

|  | OLR | RF | GRU-tp100k | GRU-tp1.6M | CNN-tp160k | CNN-tp4M |
|---|---|---|---|---|---|---|
| **Test set** | 57% | 52% | 51% | 46% | 59% | 64% |
| Train set 1 | 55% | 98% | 94% | 43% | 99% | 99% |
| Train set 2 | 63% | 100% | 100% | 100% | 100% | 100% |
| Train set 3 | 50% | 99% | 100% | 96% | 100% | 100% |
| Train set 4 | 60% | 98% | 100% | 97% | 100% | 100% |
| Train set 5 | 62% | 100% | 98% | 69% | 100% | 100% |

As expected, neural network models exhibit strong overfitting to the training set for similar reasons as the non-parametric random forest model. However, due to early stopping and dropout regularisation, they can still achieve testing set accuracy that is on par with the parametric OLR model.

The main advantage of the OLR parametric model over the neural network approach lies in the possibility to interpret intermediate steps of the classification process. This provides clinicians with more information on the recovery process, it also enables them to control and possibly adjust the automated evaluation procedure; see [13].

## 4. Discussion

The following figures show the resulting comparison of these approaches. Figure 5 represents a "theoretical" exact classification (perfect fit required), which is not so crucial for clinicians. Figure 6 shows the results of the so-called approximate classification (classifier differs from the clinician by at most 1 HB grade). This classification is most widely used in the clinical setting.

The overall comparison of the methods is then plotted in Table 5. It is evident from the overall accuracy row of the approximate classification that the OLR classification method gives the best results of all the selected approaches on these data.

In comparison of the models by HB grades, we considered two approaches: accuracy (Table 5) and precision (Table 6). The former is calculated from the row-wise distributions in a confusion matrix and it relates to most frequent approach (probability of correct classification given the true HB). The latter is calculated from the column-wise distributions in a confusion matrix and relates to the Bayesian approach (probability of correct classification given the predicted HB).

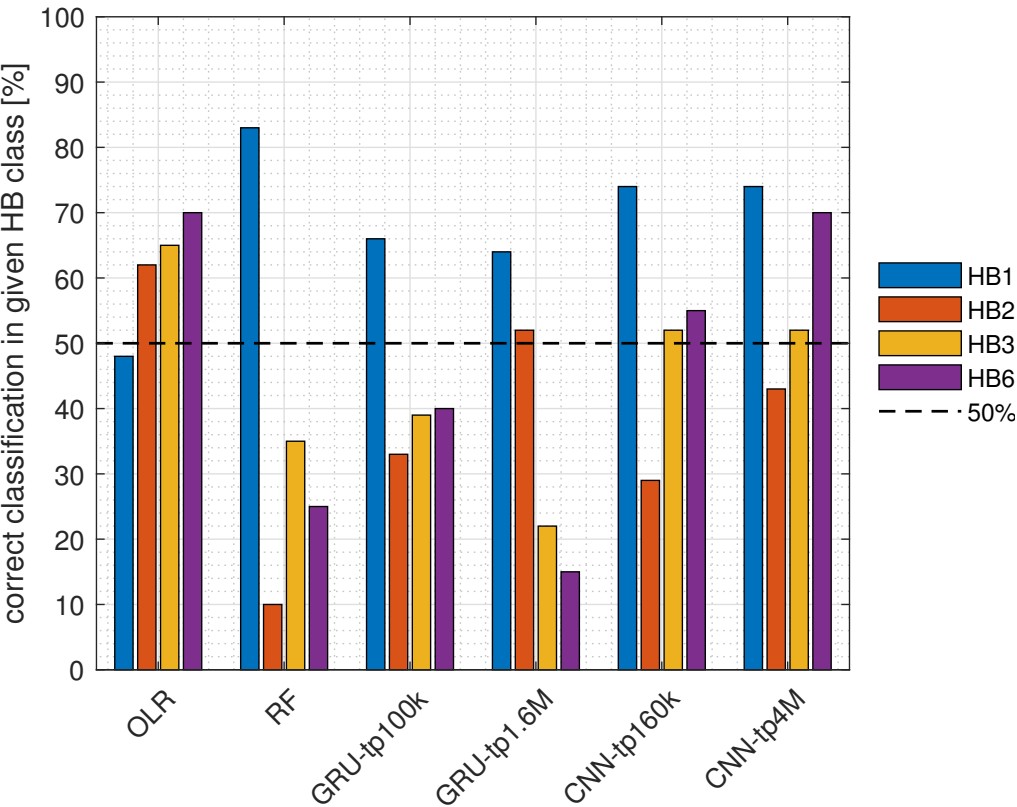

**Figure 5.** Accuracy of correct classification in different HB classes.

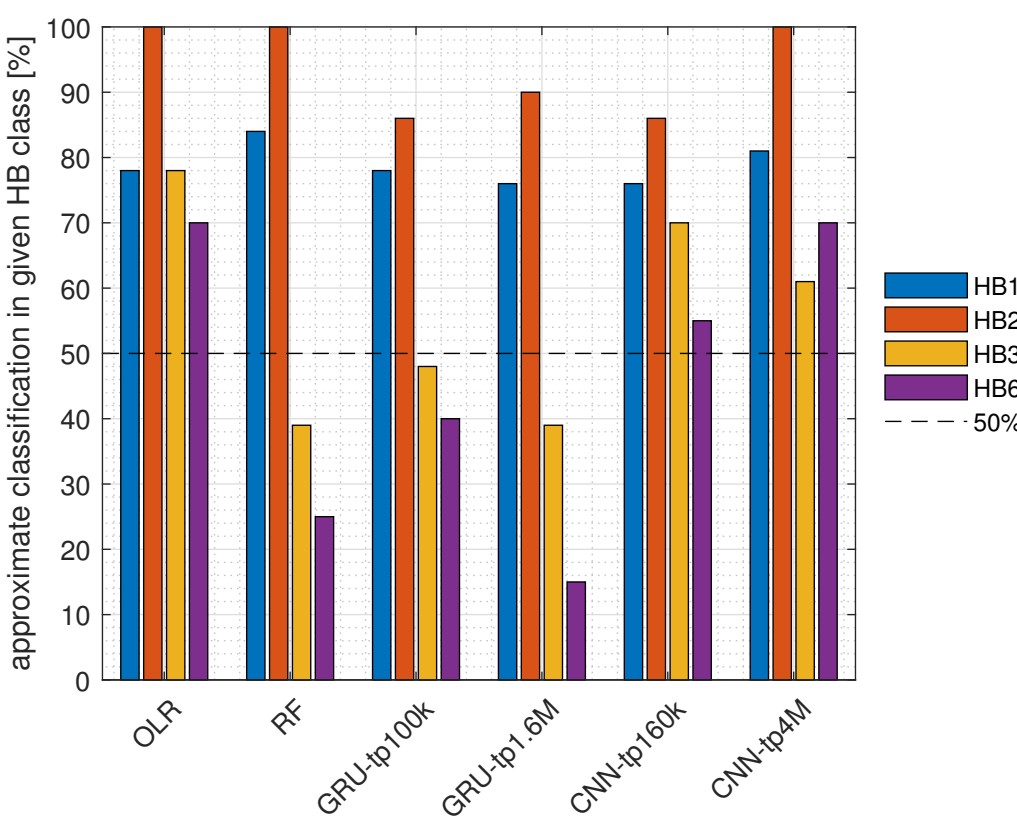

**Figure 6.** Accuracy of approximate classification in different HB classes.

## 5. Conclusions

We developed a tool for statistical analysis of mimetic movements of patients after head and neck surgery. This tool is intended for the objective assessment of the patient's rehabilitation process, which has so far only been assessed subjectively by a clinician. We compared a number of approaches:

- Parametric statistics (based on OLR)
- Non-parametric statistics (based on ordinal random forests)
- GRU
- Convolutional neural network

The parametric statistical approach based on the combination of functional logistic regression and ordinal logistic regression provided stable and reasonable results for both accuracy and precision. This demonstrates its potential to become a widely acceptable and useful method. The pros and cons of individual algorithms are summarised in Table 8.

**Table 8.** Pros and cons of observed methods.

| Method | Pros | Cons |
|---|---|---|
| statistical models in general | ✚ easily tractable, allow for analysis of underlying drivers | ➖ require careful choice of models and methods |
| parametric statistics | ✚ provide explicit dependence formulas (enable in-depth analysis of the studied phenomena) <br> ✚ do not require large datasets | ➖ sensitive to model misspecifications |
| non-parametric statistics | ✚ good flexibility <br><br> ✚ strict (distributional and model) assumptions not required | ➖ prone to overfitting on small datasets |
| neural models in general | ✚ highly modular | ➖ susceptible to sparse training data <br> ➖ black box |
| GRU based models | ✚ infer long temporal dependencies | ➖ can not reduce dimensionality, slow training/inference for longer measurement;s <br> ➖ ineffective parallelisation, longer training |
| CNN based models | ✚ infers patterns from data, both in temporal and cross-feature dimensions <br> ✚ for dimensionality reduction | ➖ more susceptible to overfitting |

Various approaches were used for data analysis, starting with standard continuous statistical data analysis and ending with convolutional neural networks. The reliability of the individual approaches was tested and compared. The resulting application is currently successfully used in the rehabilitation of patients with so-called vestibular swannoma at the Clinic of Otorinolaryngology, University hospital Královské Vinohrady, Prague.

*Future Work*

Future plans in the framework of this research focus mainly on the following

- Better data acquisition: As part of data collection and pre-processing, it is expected that the operator will be immediately informed about the quality and usability of the record;
- Unsupervised learning: In the next step, the data will be processed independently of clinical practice, which is heavily burdened by the subjective opinion of the physician;
- Fusion with the project devoted to gait analysis: The data from these experiments are fused with data from the analysis of gait of the same patients (these patients often suffer from balance problems in addition to mimic problems);
- Mobile application: In addition to a full-fledged application in a hospital environment, it is also planned to create a mobile application (using the camera system of a smartphone), which will be available to patients for home rehabilitation.

**Author Contributions:** All authors contribute equally including manuscript preparation, namely: J.K., J.M., K.Š., M.V., and J.C. software development, signal filtration and preprocessing, L.V., K.T., M.C. medical background, data acquisition, P.K., and L.Č. numerical methodology, statistical analysis. All authors have read and agreed to the published version of the manuscript.

**Funding:** The work of J.K., P.K., J.M., L.Č. was funded by the Ministry of Education, Youth and Sports by grant 'Development 406 of Advanced Computational Algorithms for evaluating post-surgery rehabilitation' number LTAIN19007 and the work of M.Ch., L.V., K.T. was funded by PROGRES Q28 Oncology, Charles University in Prague, Czech Republic. This support is gratefully acknowledged.

**Institutional Review Board Statement:** This study was conducted according to the guidelines of the Declaration of Helsinki, and it was approved by the Institutional Ethics Committee of Charles University Prague, University Hospital Kralovske Vinohrady, EK-VP/431012020, approved 22 June 2020.

**Informed Consent Statement:** Informed consent was obtained from all subjects involved in the study.

**Data Availability Statement:** The data presented in this study are available on request from the corresponding author. The data are not publicly available due to ethical restrictions.

**Acknowledgments:** We thank the Ministry of Education, Youth and Sports; Faculty Hospital Královské Vinohrady; University of Chemistry and Technology Prague.

**Conflicts of Interest:** The authors declare no conflicts of interest.

## Abbreviations

The following abbreviations are used in this manuscript:

| | |
|---|---|
| AIC | Akaike information criterion |
| CNN | Neural network with convolutional layers |
| DNN | Deep neural network |
| FLR | Functional logistic regression |
| GLM | Generalised linear models |
| GRU | Neural network with gated recurrent unit |
| HB | House–Brackmann facial nerve grading system |
| MATLAB | a proprietary multi-paradigm programming language and numerical computing environment |
| OLR | Ordinal logistic regression |
| ORL | Otorhinolaryngology |
| POI | Point of interest |
| R | a free software environment for statistical computing and graphics |
| RF | Random forests |

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
