# Peer review of "Advanced Statistical Analysis of 3D Kinect Data: A Comparison of the Classification Methods"

_applsci, doi:10.3390/app11104572_

Round 1

Reviewer 1 Report

The topic is interesting from medical and advanced engineering point of view, although thanks to the intraoperative neuromonitoring of the neural transmission during head and neck surgery, the facial paralysis happens occasionally. However, if the injury either in central nervous system or peripherally appears, the verification of mimetic muscles rehabilitation effects is difficult to establish and that is why the stereo-vision evaluation proposed by authors can be promising and its objectiveness should be verified. The authors consider clinical method of facial paresis evaluation like House-Brackman scale and their opinion about its subjectivity is right. They do not mention of precise methods of clinical neurophysiology like electromyography and electroneurography (including blink reflex study) as tools of functional analysis what is pity. My concerns refer to some medical descriptions (terms) questions included in the text.

Abstract

Line 2,3 …Work deals with a classification problem of mimetic muscle rehabilitation that is observed with by a Kinect stereo-vision camera… description is a bit confusing. Maybe the word ‘classification’ should be replaced with ‘evaluation’ problem of mimetic muscle rehabilitation throughout the text.

Introduction

  • Biomedical background

Authors present three reasons of mimetic muscles dysfunctions caused by lower motoneurone lesion and peripheral Bell’s palsy, inflammations and infections (including Herpex) and neoplasms of parotic gland influencing the paresis of facial nerve, trauma of head and neck surgery… maybe they should also recall the central reasons not caused by surgery like the ponto-cerebellar angle tumor?

  • Facial rehabilitation

The role of electrotherapy of face muscles supervised by physiotherapist and pharmacological treatment was not mentioned but it should be as a standard way of the treatment. “Stretching exercises” term is more often mentioned in medical rehabilitation literature than “retraining” or “neuromuscular re-education” terms.

  • Evaluation of facial nerve function

Authors do not mention of precise methods of clinical neurophysiology like electromyography and electroneurography (including blink reflex study) as tools of functional analysis. HB scale description should be abbreviated with citation of ref. for example House JW. Facial nerve grading systems. The Laryngoscope.  1983  Aug;93(8):1056-1069.  https://doi.org/10.1288/00005537-198308000-00016

for clarification.

2.Materials and methods

Literature provides that iatrogenic injury of face muscles motor function does not appear so often.

Authors should point out, how long data from 93 patients was collected (period of time from-to when the surgeries were performed as well as the data was collected)?

I will suggest to change the term “mimetic muscle exercises” to “mimetic muscles voluntary stretches” during evaluation with clinical HB scale.

Which face muscles were studied according to the anatomical nomenclature? Not mentioned, “mimetic” is very generalized.

2.2 Data processing

If authors modified HB scale, it should be modified in the Abstract and wider discussed in section 4 indicating the reason of modification again.

Author Response

Answer: Thank you very much for Your feedback. Professional proof-reading by Proof-Reading Service was performed.

The topic is interesting from medical and advanced engineering point of view, although thanks to the intraoperative neuromonitoring of the neural transmission during head and neck surgery, the facial paralysis happens occasionally. However, if the injury either in central nervous system or peripherally appears, the verification of mimetic muscles rehabilitation effects is difficult to establish and that is why the stereo-vision evaluation proposed by authors can be promising and its objectiveness should be verified. The authors consider clinical method of facial paresis evaluation like House-Brackman scale and their opinion about its subjectivity is right. They do not mention of precise methods of clinical neurophysiology like electromyography and electroneurography (including blink reflex study) as tools of functional analysis what is pity. My concerns refer to some medical descriptions (terms) questions included in the text.

Answer: Thank you very much for Your positive feedback.

Abstract

Line 2,3 …Work deals with a classification problem of mimetic muscle rehabilitation that is observed with by a Kinect stereo-vision camera… description is a bit confusing. Maybe the word ‘classification’ should be replaced with ‘evaluation’ problem of mimetic muscle rehabilitation throughout the text.

Answer: Thank you very much for this comment and for your careful reading. The main reason is the inconsistent terminology between the medical and engineering environments. While the term "evaluation" is commonly used in medicine, in mathematical statistics it is clearly a "classification". We went through the text and tried to correct the terminology in some places. However, for the reason described above, we have decided not to fully unify it.

Introduction

  • Biomedical background

Authors present three reasons of mimetic muscles dysfunctions caused by lower motoneurone lesion and peripheral Bell’s palsy, inflammations and infections (including Herpex) and neoplasms of parotic gland influencing the paresis of facial nerve, trauma of head and neck surgery… maybe they should also recall the central reasons not caused by surgery like the ponto-cerebellar angle tumor?

Answer: Thank you for your valuable comment. We added this paragraph to the sec. Biomedical background for better clarity.

Biomedical background  

We distinguish two major types of facial nerve palsy: central (upper) motoneuron lesion between cortex and nuclei of the facial nerve in the brainstem and peripheral (lower) motoneuron lesion between nuclei in the brainstem and peripheral organs. The most common cause of lover motoneuron lesion is idiopathic facial nerve palsy, also known as Bell’s palsy. Closely following Bell’s palsy are infection and  inflammation. Trauma, including surgical trauma in head and neck surgery, is the third  most common cause of facial nerve paralysis in the general population. Other important  etiologies of facial nerve dysfunction include herpes zoster oticus and neoplasms of the  parotid gland, brain, and the petrous part of the temporal bone (cerebellopontine angle tumors). (14)

Cases of complete paralysis after surgery in which the onset of paralysis is indeterminate should be treated as immediate in nature. Delayed paralysis or incomplete paresis should be treated medically, with high-dose steroids. A good prognosis should be anticipated in these cases. (6)

  • Facial rehabilitation

The role of electrotherapy of face muscles supervised by physiotherapist and pharmacological treatment was not mentioned but it should be as a standard way of the treatment. 

Answer:  Thank you for this comment. We added two paragraphs to the sec. Facial rehabilitation.

Pharmacological therapy  

The pharmacological treatment depends on the cause of facial nerve paresis and generally most often are used corticosteroids, antiviral agents, calcium channel blockers, vitamins to support regeneration of the nerve. 

Neuromuscular electrical stimulation  

Neuromuscular electrostimulation therapy is used for the direct or indirect therapeutic stimulation of nerves, muscles. In general, three stimulation frequency ranges can be distinguished: low-frequency, medium-frequency and high-frequency currents. The low-frequency stimulation currents (up to about 1000 Hz) are suitable for creating synchronous muscle contractions. Stimulation can be sensory or motor, but it should always be below the discomfort threshold. The current is applied via electrodes. These can either be inserted into the tissue surrounding a nerve, or into the muscle (percutaneous stimulation) or into the skin applied (transcutaneous stimulation). (5)

“Stretching exercises” term is more often mentioned in medical rehabilitation literature than “retraining” or “neuromuscular re-education” terms.

Answer: Thank you very much for this point. Based on your proposal, we have unified the terminology in the text (sec. Facial rehabilitation).

  • Evaluation of facial nerve function

Authors do not mention of precise methods of clinical neurophysiology like electromyography and electroneurography (including blink reflex study) as tools of functional analysis. 

Answer:  Thank you once more for this comment, we added these paragraphs to the manuscript (sec. Electrodiagnostics of facial nerve palsy).

Electrodiagnostics of facial nerve palsy  

Electrophysiological tests are mainly used to determine the severity and prognosis of a peripheral facial nerve lesion. (1) 

Electroneurography objectively records the amplitude of electrically evoked muscle action potentials. Analyzes the evoked compound muscle action potential (CMAP) of a specific facial muscle after transcutaneous stimulation of the main trunk of the facial nerve. (2) The main trunk is stimulated supramaximally at its exit from the stylomastoid foramen with a bipolar stimulator or stimulating electrodes. The CMAP is recorded using a bipolar pair of surface electrodes placed on the target muscle. Important test between 72 h and 21 days after onset, interpretation of result in comparison to nEMG (needle electromyography) result. (3) Nerve injury is expressed as percentage of function relative to the normal side. (6) 

Electromyography (EMG) measures volutional responses of the facial muscles without electrostimulation. A facial motoneuron unit consists of a facial motoneuron and all muscle fibers innervated by this motoneuron. Needle EMG (nEMG) is the method used to analyze a facial motor unit action potential (MUAP) recorded from a needle electrode inserted in the facial muscle. This examination is  important 2–3 weeks after onset of the palsy, because pathologic activity can occur in case of facial nerve degeneration. In the later time course, nEMG is important to detect reinnervation potentials as signs of reinnervation of the facial muscles (2). Surface electromyography (sEMG) works with voluntary activity of the facial muscles and not with external stimulation. The recording field  is more superficial and larger than when using nEMG.  sEMG is not used for prognostication. Multichannel sEMG is important if the interplay of different facial muscles should be analyzed.  

Blink-reflex testing is a test that allows stimulation of the facial nerve proximal to the lesion site. Testing is electrostimulation of the supraorbital branch of the trigeminal nerve (V1) and simultaneous sEMG recording from the orbicularis oculi muscle on both sides. Standard blink testing involves electrical stimulation of the supraorbital nerve on the affected side combined with a 2-channel simultaneous sEMG recording from both orbicularis oculi muscles. The exit of the supraorbital nerve in the supraorbital foramen is palpated on the rim of the orbit. (4) It may be most helpful if facial nerve damage is suspected to occur within the brainstem. (2) 

HB scale description should be abbreviated with citation of ref. for example House JW. Facial nerve grading systems. The Laryngoscope.  1983  Aug;93(8):1056-1069.  https://doi.org/10.1288/00005537-198308000-00016 for clarification

Answer: We corrected it according to your suggestion.

2.Materials and methods

Literature provides that iatrogenic injury of face muscles motor function does not appear so often.

Answer: Thank you for the comment. According to [Hohman MH, Bhama PK, Hadlock TA. Epidemiology of iatrogenic facial nerve injury: a decade of experience. Laryngoscope. 2014 Jan;124(1):260-5. doi: 10.1002/lary.24117. Epub 2013 Apr 18. PMID: 23606475. ] “Iatrogenic facial nerve injury occurs most commonly in temporomandibular joint replacement, mastoidectomy, and parotidectomy where the risk is resulting in a high incidence of at least temporary facial weakness: up to 40%, because of the manipulation of facial nerve. Direct visualization of the nerve and facial monitoring may decrease the incidence of injury, and early referral for facial nerve exploration may result in improved outcomes.” 

Authors should point out, how long data from 93 patients was collected (period of time from-to when the surgeries were performed as well as the data was collected)?

Answer:  Thank you for the comment. The data (processed in the article) were collected from 3.2.2019 to 14.5.2020. At the beginning of the project, it was mainly test data for debugging and verification of the correct functionality of the system, then the data were collected at a much higher frequency.

I will suggest to change the term “mimetic muscle exercises” to “mimetic muscles voluntary stretches” during evaluation with clinical HB scale.

Answer: Thank you very much for this point, based on your proposal, we have unified the terminology in the text.

Which face muscles were studied according to the anatomical nomenclature? Not mentioned, “mimetic” is very generalized.

Answer: Thank you for this point. We checked the text and added this paragraph: 

The muscles of the face include all mimetic muscles innervated by the cranial nerve VII (facial nerve).  Within the parotid gland, the facial nerve terminates by bifurcating into five motor branches. These innervate the muscles: temporal branch –  innervates the frontalis, orbicularis oculi and corrugator supercilii, zygomatic branch – innervates the orbicularis oculi, buccal branch – innervates the orbicularis oris, buccinator and zygomaticus muscles, marginal mandibular branch – innervates the mentalis muscle.cervical branch – innervates the platysma. Two masticatory muscles (masseter, temporalis) that are supplied by the motoric portion of the cranial nerve V3 (mandibular nerve) also contribute to the contour of the face. (8.) 

2.2 Data processing

If authors modified HB scale, it should be modified in the Abstract and wider discussed in section 4 indicating the reason of modification again.

Answer: We added it directly into abstract. Thank you for this note.

  1.  Thielker J, Grosheva M, Ihrler S, Wittig A, Guntinas-Lichius O. Contemporary management of benign and malignant parotid tumors. Front Surg. 2018;5:39.  
  2. Guntinas-Lichius O, Volk GF, Olsen KD, Mäkitie AA, Silver CE, Zafereo ME, Rinaldo A, Randolph GW, Simo R, Shaha AR, Vander Poorten V, Ferlito A. Facial nerve electrodiagnostics for patients with facial palsy: a clinical practice guideline. Eur Arch Otorhinolaryngol. 2020 Jul;277(7):1855-1874. doi: 10.1007/s00405-020-05949-1. Epub 2020 Apr 8. PMID: 32270328; PMCID: PMC7286870. 
  3. Heckmann JG, Urban PP, Pitz S, Guntinas-Lichius O, Gagyor I. The diagnosis and treatment of idiopathic facial paresis (bell's palsy) Dtsch Arztebl Int. 2019;116:692–702. 
  4. Kennelly KD. Electrodiagnostic approach to cranial neuropathies. Neurol Clin. 2012;30:661–684. 
  5. Miller S, Kühn D, Jungheim M, Schwemmle C, Ptok M. Neuromuskuläre Elektrostimulationsverfahren in der HNO-Heilkunde [Neuromuscular electric stimulation therapy in otorhinolaryngology]. HNO. 2014 Feb;62(2):131-8; quiz 139-40. German. doi: 10.1007/s00106-013-2810-4. PMID: 24549514. 
  6. Gordin E, Lee TS, Ducic Y, Arnaoutakis D. Facial nerve trauma: evaluation and considerations in management. Craniomaxillofac Trauma Reconstr. 2015 Mar;8(1):1-13. doi: 10.1055/s-0034-1372522. PMID: 25709748; PMCID: PMC4329040. 
  7. Hohman MH, Bhama PK, Hadlock TA. Epidemiology of iatrogenic facial nerve injury: a decade of experience. Laryngoscope. 2014 Jan;124(1):260-5. doi: 10.1002/lary.24117. Epub 2013 Apr 18. PMID: 23606475. 
  8. von Arx T, Nakashima MJ, Lozanoff S. The Face – A Musculoskeletal Perspective. A literature review. Swiss Dent J. 2018 Sep 10;128(9):678-688. PMID: 30056693.

Reviewer 2 Report

Thank you very much for your hard work in performing this very interesting study. Your study will have clinical significance for classification problem of mimetic muscle rehabilitation in facial paresis patients following head and neck surgery.
-Although parametric statistics generally outperform nonparametric statistics, many studies have implemented nonparametric statistics. What is the specific reason for comparing nonparametric and parametric statistics in this study?
-After head and neck surgery, the mechanisms of facial paresis should be described in detail

Author Response

Answer: Thank you very much for Your feedback. Professional proof-reading by Proof-Reading Service was performed.

Thank you very much for your hard work in performing this very interesting study. Your study will have clinical significance for classification problem of mimetic muscle rehabilitation in facial paresis patients following head and neck surgery.

Answer: Thank you very much for Your positive feedback.

Although parametric statistics generally outperform nonparametric statistics, many studies have implemented nonparametric statistics. What is the specific reason for comparing nonparametric and parametric statistics in this study?

Answer: Thank you very much for this comment. On one hand, we deal with the classification problem with very complex explanatory data (collection of curves for each observation)  having unclear functional link to the response variable, which suggests using nonparametric statistics (or neural networks) as a preferred option. On the other hand, the number of observations is very limited, which increases the risk of overfitting for these methods. Therefore we decided to compare them with parametric statistics, which is less prone to overfitting, but requires very careful model specification to get reasonable results. (added into sec. Compared statistics)

-After head and neck surgery, the mechanisms of facial paresis should be described in detail

Answer:  Thank you for this point and for careful reading. We extended the sec. Biomedical background and added a reference to our previous publication where this mechanism is described in more details.